# COVID-19, Anosmia, and Allergies: Is There a Relationship? A Pediatric Perspective

**DOI:** 10.3390/jcm11175019

**Published:** 2022-08-26

**Authors:** Giulia Brindisi, Alberto Spalice, Caterina Anania, Flaminia Bonci, Alessandra Gori, Martina Capponi, Bianca Cinicola, Giovanna De Castro, Ivana Martinelli, Federica Pulvirenti, Luigi Matera, Enrica Mancino, Cristiana Alessia Guido, Anna Maria Zicari

**Affiliations:** 1Department of Mother-Child, Urological Science, Sapienza University of Rome, 00161 Rome, Italy; 2Department of Translational and Precision Medicine, Sapienza University of Rome, 00185 Rome, Italy; 3Department of Molecular Medicine, Sapienza University of Rome, 00161 Rome, Italy; 4Primary Immune Deficiencies Unit, Department of Internal Medicine and Infectious Diseases, Azienda Ospedaliera Universitaria Policlinico Umberto I, 00185 Rome, Italy

**Keywords:** allergies, COVID-19, anosmia, SARS-CoV-2 infection, pandemic, body mass index, disease duration, mean nasal flow, active anterior rhinomanometry, pediatric

## Abstract

Background: Between June and July 2020, we evaluated children and adolescents concerning post-infection surveillance after a COVID-19 positivity during the lockdown. We aimed to assess whether the anamnestic presence of allergies could correlate with the presence of SARS-CoV-2 symptoms, and in particular with anosmia. Material and methods: For each patient, we collected anamnestic data, the presence of allergies documented by performing skin prick tests, and COVID-19 symptoms. Then, if over six years of age, each patient underwent an active anterior rhinomanometry. Results: A total of 296 patients were enrolled, of whom 105 (35.4%) reported allergies. Considering COVID-19 symptoms, 74 subjects (25%) presented an asymptomatic form, 222 (75%) reported symptoms, and anosmia recurred in 60 subjects (27.03%). A statistically significant relationship was found between allergies and symptomatic COVID-19 (*p* = 0.042), allergies, and anosmia (*p* = 0.05), and allergies and anosmia in males (*p* = 0.007). Moreover, anosmic patients presented a higher body mass index, older age, and a longer COVID-19 duration with statistical significance (*p* = 0.001, 0.001, 0.006, respectively). Conclusions: Allergic subjects seem to develop symptomatic COVID-19 more frequently and allergies appear to be a protective factor from anosmia’s onset in males.

## 1. Introduction

The outbreak of novel coronavirus disease 2019 (COVID-19) due to the “severe acute respiratory syndrome coronavirus 2 (SARS-CoV-2)” radically changed the way we live, work, and relate to other people for about two years now. The first period following the outbreak of the pandemic put a strain on the whole world from a medical, social, moral, and economic point of view. Today, after an alternation of longer and shorter lockdown or semi-lockdown periods all over the world and the start of vaccination campaigns, the situation has changed dramatically. It can be said that we have almost returned to pre-pandemic life, while attention remains high on preventing the viral spread which is nevertheless still elevated in various areas of the world [1].

During the pandemic, the way of approaching chronic pathologies in the pediatric age also changed. The use of surveys, as well as direct contacts via phone calls or emails, was very useful for monitoring symptoms during the lockdown period [2,3].

However, these methods have not been able to prevent the negative effects of long lockdown periods on the psyche of children and adolescents as well as on their bodies, with a registered increase in overweight or obesity, compared to the pre-pandemic era [4,5].

COVID-19, declared a pandemic in March 2020 by the World Health Organization (WHO), caused millions of deaths all over the world [6,7].

Children are almost always asymptomatic or paucisymptomatic. When symptoms occur, COVID-19 in pediatric age is characterized by fever, skin or gastrointestinal manifestations, nasal congestion, pharyngodynia, arthomyalgia, cough, and pneumonia, up to cases of respiratory distress that require hospitalization in intensive care. Symptoms such as the loss of smell (anosmia) and taste (ageusia), the most characteristic symptoms of COVID-19, are more challenging to diagnose in children than in adults. This difficulty is linked to the inability of younger children to report these symptoms and the lack of valid diagnostic tests to evaluate pediatric anosmia and ageusia [8,9,10].

Recently, the term Long COVID-19 syndrome (LCS) was coined to describe the condition that occurs in people with a history of positive SARS-CoV-2 infection, with one or more persistent physical symptoms for a minimum duration of 12 weeks after the initial positive test for SARS-CoV-2; these symptoms cannot be explained by an alternative diagnosis [11,12].

These symptoms may continue uninterrupted or begin as relapses after a short latency period. It has even been seen that this phase is more serious for subjects who presented a mild initial phase of COVID-19 [13].

Not only in the acute phase of the disease but also in the post-acute phase, chemosensitive symptoms such as anosmia and ageusia are the most frequently reported symptoms among patients [14].

Allergic rhinitis (AR) and asthma are the most common allergic diseases that very often, due to the anatomical continuity between the upper and lower airways and their reciprocal crosstalk, are considered a single disease as the theory “one airway one disease” explains [15]. In fact, they both share the same type of T helper 2 (Th2) inflammation and several biomarkers responsible for the chronic clinical manifestations [16].

Symptoms of AR and asthma can overlap with those of COVID-19, so careful monitoring of COVID-19 symptoms among allergy sufferers is essential to avoid the viral spread. In particular, anosmia is one of those symptoms that is characteristically shared by AR and COVID-19 [17].

Regarding the role of allergic sensitization and COVID-19, it has been shown that allergies are not a risk factor for contracting SARS-CoV-2 infection. One of the first studies conducted by Du H et al., analyzing 182 children admitted to hospital for COVID-19, showed the absence of statistically significant differences in disease severity and development of complications between allergic and non-allergic subjects [18].

Additionally, the recent surveillance study conducted by Seibold MA et al. to determine the incidence of SARS-CoV-2 infection in children and adults affected by asthma and/or other allergic diseases in the family context, showed that allergies did not increase the risk of SARS-CoV-2 infection [19].

However, uncontrolled allergic symptoms constitute a marker of a poor disease prognosis [20]. Therefore, especially during the pandemic and in the case of SARS-CoV-2 infection, it is necessary to achieve allergic symptom control [21].

Therefore, considering that a controlled allergic disease is not a susceptibility marker to contract SARS-CoV-2, the primary aim was to evaluate the relationship between allergies and the development of a symptomatic infection among a cohort of pediatric patients, all tested positive at nasal swab during the lockdown, and assessed as part of a post-infection surveillance program. In addition, another aim of this study was to detect the role of the allergic condition concerning anosmia, one of the most important and distinctive COVID-19 symptoms.

To the best of our knowledge, there are no updated data that verified at the same time the role of allergies in both the development of symptoms during COVID-19 and anosmia among a pediatric population.

## 2. Materials and Methods

Between June and July 2020, in the period following the first lockdown, which in Italy lasted from the beginning of March until the end of May 2020, we evaluatedm at the Pediatric Allergy Unit of the Department of Mother Child Urological Science, Sapienza University of Rome, children and adolescents who all had experienced COVID-19 in the lockdown period as part of a post-infection surveillance program. This pilot study was performed according to the Declaration of Helsinki regarding biomedical research involving human subjects, and the study protocol was approved by the local ethics committee of Sapienza University of Rome (number protocol 0399/2021). Written informed consent was obtained from the parents or legal tutors of all the enrolled patients. The main exclusion criterion was the lack of willingness to be included in the study, the lack of anamnestic data useful for the subsequent statistical analysis of the data, and the possible detection of the presence of self-reported anosmia already present before COVID-19. Specifically, we investigated the presence of new-onset anosmia excluding all patients (SARS-CoV-2 positive with or with allergies) with a pre-existing self-reported olfactory dysfunction, a history of chronic nasal/sinus infections, or a history of endoscopic sinus surgery, in addition to neuropsychiatric and syndromic pathologies that may compromise olfactory function. Diabetes mellitus, renal impairment, and any other pathology that interferes with olfactory function were also included. 

Expert doctors carried out a careful physical examination with accurate anamnestic history. Then for each patient, were collected data on gender, anthropometric parameters (weight, height, body mass index (BMI)), presence of allergies (AR, asthma, atopic dermatitis, conjunctivitis, urticaria) documented by performing skin prick tests (SPTs), and COVID-19 symptoms. We only enrolled patients with controlled allergic disease and evaluated them through Asthma Control Test (ACT) [15], for the assessment of asthma control and the visual analog scale (VAS) for AR control [22]. Then, if over six years of age, each patient underwent an active anterior rhinomanometry (AAR).

### 2.1. Skin Prick Tests (SPTs)

We used a complete panel for aeroallergens and food allergens (ALK-Abelló, Hørsholm, Denmark). As positive and negative controls, we used histamine dihydrochloride 10 mg/mL and glycerol saline solution, respectively. To prick the skin, we adopted Morrow-Brown needles and after 15 min wheal reactions were analyzed, considering as positive a wheal ≥ 3 mm [23].

### 2.2. COVID-19 Symptoms

With regard to SARS-CoV-2 infection, it was asked whether the disease occurred with or without symptoms and, if they were present, which ones. All COVID-19 symptoms were self-reported by patients and supported by parents at the time of the visit. The duration of the disease (number of positive days on the nasal swab test) was also investigated, as well as the possible need for hospitalization.

### 2.3. Anterior Active Rhinomanometry (AAR)

All the eligible patients over six years of age underwent an AAR to evaluate the nasal flow in each nostril [24]. This exam allows the objective determination of nasal flow according to ICSR (Committee for the Standardization of Rhinomanometry) [25], using a RINOPOOCKET ED200 (EUROCLINIC^®^, Imola, Bologna, Italy). Nasal obstruction is classified as reported below: very severe if the airflow values are less than 29% of the expected, severe if the values are between 29% and 42%, moderate if the values are between 43% and 56%, mild for values between 57% and 70% and absent for values above 71% [24,25].

#### Statistical Analysis

Statistical analysis was performed using IBM SPSS version 27.0 (SPSS, Chicago, IL, USA). Normality tests were made for continuous variables. Each of them was represented by mean value and standard deviation (SD). The nominal variables were represented in terms of relative counts and frequencies. The existence of relationships between the nominal variables was investigated with the chi-square test and the statistical significance was evaluated at the threshold of 0.05. In the cases of statistically significant relationships, the risk assessment was carried out by odd ratio (OR) and relative confidence interval (CI) of 95%. Some descriptive parameters characteristic of the sample were analyzed based on the presence or absence of the variables of interest. The mean values of these groups were compared using Student’s *t*-test. Finally, a logistic regression model was built which could predict the onset or not of anosmia as a function of six covariates: gender, BMI, age, duration, mean nasal flow (mNF), and allergies.

## 3. Results

A total of 296 patients, all of whom tested positive for SARS-CoV-2 infection during the lockdown, agreed to be enrolled in this study. The characteristics of the study population are reported in Table 1 below.

Allergies were present in 105 patients (35.47%); detailed description of allergic symptoms is reported in Table 2.

All the enrolled patients had SARS-CoV-2 infection during the lockdown period. Among them, only 74 subjects presented an asymptomatic form of infection (41 females; 33 males), 222 had one or more COVID-19 symptoms (99 females; 123 males); in particular, among the latter, 54 (24.32%) reported just one symptom, while 168 (75.68%) more than one. Among all the COVID-19 symptoms, anosmia recurred in 60 patients (27.03%), among these were 31 females and 29 males.

The COVID-19 symptoms reported by patients during the lockdown period are described in Table 3. The mean duration of the COVID-19 was 20.09 ± 8.05 days. None required hospitalization and/or intensive care admission.

After the descriptive analysis of the data, we investigated the existence or not of a relationship between allergies and the presence of symptoms during SARS-CoV-2 infection (Table 4). To do this, we used a chi-square test (X^2^ Pearson = 4.138) to highlight a relationship between allergies and COVID-19 symptomatology with statistical significance (*p* = 0.042). The evaluation of the estimated risk (OR = 1.83; CI 95%: 1.017–3.293) attributes the value of a risk factor to the pre-existence of allergies. From this, it seems that not only is there a statistically significant relationship between allergies and the development of symptomatic COVID-19 disease but also that allergies constitute a risk factor for the development of COVID-19 symptoms.

Subsequently, the existence or not of a relationship between allergies and anosmia was investigated, as shown in Table 5. A chi-square test (X^2^ Pearson = 3.606) showed a statistically significant relationship between allergies and anosmia (*p* = 0.05). This association is statistically significant in subjects over 5 years of age (*p* < 0.027).

Specifically, by stratifying the sample by gender, there has been a confirmation of the existence of a statistically significant relationship between allergies and anosmia in the male group (Female: X^2^ Pearson = 0.013, *p* = 0.91; Male: X^2^ Pearson = 7.149, *p* = 0.007), as shown in Table 6. The subsequent evaluation of the estimated risk attributes to the pre-existence of allergies in the male group, the value of a protective factor from the development of anosmia with statistical significance (Females OR = 1.050; CI 95% 0.447–2.470; Males OR = 0.264; CI 95% 0.095–0.736) (Table 6).

Furthermore, COVID-19 related anosmia was investigated in relation to some characteristics of our population or the disease.

For this purpose, regarding each variable considered, we determined the average values in the two groups: with or without anosmia. These average values were compared by employing of a Student’s *t*-test, which highlighted what is reported in Table 7.

The data in the table show that anosmic patients presented a higher body mass index (BMI), older age, and a longer COVID-19 disease duration with statistically significant values.

As the final step in our analysis, a logistic regression analysis was performed to assess the ability of a model based on the variables gender, BMI, allergies, COVID-19 disease duration, flow, and age to predict the onset or not of COVID-19-related anosmia.

In other words, we applied a logistic regression model with the response variable Y (anosmia) = 1 (yes) or 0 (not) and covariates gender, BMI, allergies, COVID-19 disease duration, flow, and age (Table 8). The obtained confusion matrix shows the observed anosmia values and the predicted ones by the model, as well as the percentage of the correctness of the model. It can be seen that the model, as constructed, can correctly predict the absence of anosmia equal to 96.3% and an ability to correctly predict the onset of anosmia equal to 30%. However, the model shows a good overall predictive capacity equal to 81.9%. We reported, for each covariate of the model, the Wald test for the significance of the individual covariates and the value of OR (Exp (B)) associated with each of them with the relative CI of 95% (Table 8). From this analysis, it follows that the covariates BMI, allergies, duration, and age are statistically significant, while flow and gender are not. All the significant covariates are propaedeutic to the correct prediction of the onset of anosmia. Among these, the one that contributes most to increasing this predictive capacity is allergies.

## 4. Discussion

We analyzed a wide cohort of children and adolescents, all of whom had tested nasal swab positive during the lockdown and were evaluated in our department as part of a post-infection surveillance program.

Analysis of our results showed that: (1) allergic subjects seem to develop more frequently a symptomatic SARS-CoV-2 infection rather than an asymptomatic form; (2) the presence of a relationship between allergies and anosmia; moreover, allergies appear to be a protective factor against the onset of anosmia among males; (3) anosmic patients presented a higher BMI, older age and longer COVID-19 duration.

To the best of our knowledge, this is the first comprehensive analysis investigating the same time in a pediatric population affected by COVID-19 all these correlations: allergies and symptomatic COVID-19, allergies, and anosmia and anosmia concerning BMI, age, and duration of COVID-19.

### 4.1. Allergy and Symptomatic SARS-CoV-2 Infection

Data from the recent literature on this topic, show that allergies do not increase the risk to contract SARS-CoV-2 infection [17,18,19,20,21]. There is more than one explanation. Allergic patients present a characteristic Th2 inflammation and quite often a picture of eosinophilia in their blood count. These features, as reported by Dong et al., seem to exert a positive role in COVID-19 pathogenesis [26]. In particular, the higher the eosinophilic count is, the better the prognosis of COVID-19 with reduced morbidity and mortality [27].

Another noteworthy point is that allergic subjects present a reduced expression of the angiotensin-converting enzyme receptor 2 (ACE2), the known gateway of SARS-CoV-2 in upper airway cells. The expression of ACE2 is down-regulated by the Th2 response, in particular by IL-4 and IL-13, exerting a protective role against SARS-CoV-2 [28,29].

On the other hand, Th1 response, primarily with interferon-gamma (IFN-gamma), increases the expression of these receptors, enhancing SARS-CoV-2 entry into cells. Therefore, the Th1- Th2 balance is implicated in COVID-19 pathogenesis and the defense against SARS-CoV-2 [30]. The third point is that asthma drugs, such as inhaled steroids, bronchodilators, and montelukast, seem to be able to inhibit the inflammation caused by SARS-CoV-2 [31].

However, although allergies do not represent a predisposing factor for contracting COVID-19 or an aggravating one in terms of hospitalization or life risk [6,32] to date, the present authors focused their attention more on the susceptibility to the infection and its severity among allergic patients. Indeed, our idea was to detect in the pediatric age how allergic sensitization impacts on development of a symptomatic infection rather than an asymptomatic one. Investigating this, our findings show that in the case of SARS-CoV-2 infection among allergic subjects a symptomatic disease develops more easily rather than an asymptomatic infection.

One of the first studies at the beginning of the pandemic, conducted by Du H et al., analyzing 182 children admitted to hospital for COVID-19, found that there were no statistically significant differences in disease severity and the development of complications between allergic and non-allergic patients [18].

Additionally, Vezir Emine et al. evaluated from 1 to 3 months after discharge 75 subjects (5–18 years of age) all affected by SARS-CoV-2 infection and admitted to the pediatric emergency room by performing total IgE, SPTs, and spirometry. The authors found that AR and asthma were associated with a milder course of SARS-CoV-2 infection [33].

Similar findings were described by L.E. Eggert and co-workers, in a wide cohort of asthmatic patients (6.976) who tested positive for SARS-CoV-2, finding that allergic asthma reduced the risk of hospitalization with a protective effect compared with non-allergic asthma (OR 0.52 [0.28–0.91], *p* = 0.026) [34]. Therefore, they pointed out how asthma and allergic sensitization were not risk factors for more severe COVID-19, and did not increase the risk of hospitalization [34].

Recently, the Global Asthma Network published an article in which the clinical manifestations of SARS-CoV-2 infection were evaluated among 169 children with asthma enrolled in 10 countries. The authors found that only 58 patients (34.3%) were asymptomatic, while the majority had the symptomatic form of COVID-19 with 93 patients (55%) presenting mild symptoms, 14 (8.3%) moderate, and 4 (2.4%) severe symptoms, and among those with moderate or severe symptoms, greater exacerbations of asthma were reported [35].

However, as explained above, very extensive literature to date has been published explaining the role of allergies as a protective factor against the susceptibility to contract COVID-19 and in the case of acute infection against the development of severe COVID-19 [26,27,28,29,30,31].

In addition, since the role of “Long COVID-19 Syndrome” (LCS) has recently been brought to the attention of the scientific community, there has consequently been interested in the relationship between this LCS and allergies. Several studies have already reported that allergies seem to be associated with a higher risk of persistent symptoms during the follow-up period post-infection [13,36,37].

All of these published data agree that the presence of allergies is not relevant to acquiring a more severe case of COVID-19. Our results add to these observations that the allergic phenotype instead becomes relevant in the development of symptoms, although not severe, during SARS-CoV-2 infection.

### 4.2. Allergy and Anosmia

Considering anosmia, several studies reported this symptom as one of the most remarkable and disease-specific findings during both the acute phase of COVID-19 and the post-acute phase (LCS), although data on its prevalence varies from country to country [3,6,9,10].

A multicenter prospective cohort study conducted by Aysegul Elvan-Tuz and coworkers in Turkey analyzed COVID-19-related anosmia in a cohort of 10.157 patients aged between 10–18 years of age, and anosmia was found in 12.5% (1.266) of COVID-19 cases [38]. Another study among 141 patients (10–19 years of age) reported anosmia with a prevalence of 24.1% (34 patients) [39].

In a recent prospective clinical cross-sectional study, conducted on 79 children with COVID-19, 86.1% of them reported smell impairment. Olfactory dysfunction, in this study, was considered an early and common symptom that could be overcome by the end of the first month after the infection in the majority of patients (94.3%) [40]. The online survey elaborated by Saad N. Algahtani and co-workers, in a cohort of 881 adults, found that anosmia during the acute phase was reported in 72% of the enrolled patients and it persisted also during the post-acute phase in 33.8% of cases [14].

Additionally, in our study, anosmia was one of the most prevalent COVID-19 symptoms (27.03%) among all the enrolled symptomatic patients.

Moreover, considering other factors, in the literature COVID-19 related anosmia seems to be more linked to females [14,41,42]; in our population females seem to be more prone to develop anosmia than males, although these findings were not supported by a statistical significance. Nevertheless, this aspect remarks the importance to investigate gender differences in the clinical expression of COVID-19 in further studies [14,41,42].

However, to the best of our knowledge, none of the already published studies evaluated the role of the relationship between COVID-19 related anosmia and allergies in a pediatric population. The preliminary results of our study showed a statistically significant relationship between anosmia and allergies. In addition, an interesting and new finding comes from the analysis carried out by stratifying according to gender: being male as well as allergic resulted in a protective factor against the development of COVID-19-related anosmia. This is in line with the previously explained link between the female gender and the development of anosmia reported in the literature. However, to date, nothing has been said yet about the protective role of the male gender, particularly when associated with an allergic phenotype, toward the development of COVID-19-related anosmia.

### 4.3. Anosmia vs. BMI, Age, and COVID-19 Duration

Furthermore, as highlighted in Section 3, we found that anosmia was related not only to the gender variable but also to other factors. In particular, in our study, anosmia seemed to be related, with statistical significance to a higher BMI, older patients’ age, and longer COVID-19 duration. Conversely, no correlation was found with other factors considered such as the mNF evaluated at AAR.

Considering BMI, it has already widely been described the statistically significant correlation between “non COVID-19 related anosmia” and higher BMI values [43,44]. Instead, there is little data available on the loss of olfactory senses in COVID-19 patients with a high BMI (overweight or obese). An interesting review by Amira Sayed Khan underlined that obesity represents a large risk factor for SARS-CoV-2 infection. In fact, it can hide anosmia often present in both the obese phenotype and COVID-19, leading to a higher mortality rate [45]. In addition, obese patients with more extensive adipose tissue where ACE2 receptors are expressed are more susceptible to the development of COVID-19 infection itself [5,45,46,47].

The survey presented by Saad N. Algahtani et al. showed that anosmia and ageusia were the most prevalent symptoms during both the acute as well as post-acute phases among 881 adults affected by COVID-19. The multivariable analysis highlighted that anosmia was significantly associated with BMI, asthma, and shortness of breath during the acute phase, while with gender (females) during the post-acute phase (LCS) [14].

To the best of our knowledge, regarding the correlation between anosmia and age, no studies analyzed this relationship among a pediatric population affected by SARS-CoV-2 infection.

Studies on adults found that anosmia affected more young adults (between 20–40 years of age) in comparison with older adults, tending also to be associated with a longer persistence of anosmia itself in the post-acute phase [14,42,48].

Finally, to the best of our knowledge, no reports are available to date on the correlation between anosmia and the duration of the acute phase of disease among a pediatric population. Instead, further data are available in adulthood on the persistence and duration of anosmia in the post-acute phase of the disease, being anosmia one of the most frequently reported and studied symptoms in LCS [41,42,48,49].

In conclusion, no studies to date have considered the use of AAR in pediatric patients in post-infection follow-up control. Unfortunately, no statistically significant nasal flow changes at AAR were found in those who reported anosmia in the acute COVID-19 phase. It would be desirable to carry out this examination in pediatric patients with the persistence of anosmia in the post-acute phase (LCS).

### 4.4. Limits

Our results should be interpreted considering some limitations. First of all, this is a pilot study related only to a single center in Rome. Moreover, it was a retrospective investigation as patients were examined in the post COVID-19 lockdown, within a post-infection surveillance program. Lastly, anosmia was considered a reported symptom at the time of the follow-up visit and was not objectified during the acute phase of infection. Several questions remain unsolved, so further studies, in more than one setting, are warranted to confirm these results in a long-lasting follow-up. Only by extending this approach to other Italian pediatric centers that have followed children and adolescents who have contracted SARS-CoV-2 infection, we could reinforce our findings.

## 5. Conclusions

Allergic subjects when infected seem to develop a form of COVID-19 that is symptomatic rather than asymptomatic. Furthermore, in the pediatric population, considering anosmia which is one of the cardinal symptoms of COVID-19, it was seen that being male and allergic seems to be a protective factor against the development of this symptom during infection. Then it appears that anosmic patients presented a higher BMI, an older age as well as a longer duration of COVID-19 in the acute phase. COVID-19 is a new disease, thus we expect that more studies should be carried out to improve actual knowledge over time. In addition, in the era of the allergy epidemic and the COVID-19 pandemic, anosmia should be more investigated especially in the pediatric age where this symptom is mostly reported, given the lack of standardized tests that can objectify it. Moreover, the burden of persistent and undetected anosmia can be relevant for future physical and cognitive development.

## Figures and Tables

**Table 1 jcm-11-05019-t001:** Characteristics of the study population related to all the enrolled patients.

Characteristic	Value (*)
**No. patients**, *n* (%)	296 (100)
* Female*	140 (47.30)
* Male*	156 (52.70)
**Age**, *mean* ± *SD*, *years*	9.86 ± 4.20
* Female*	10.02 ± 4.24
* Male*	9.72 ± 4.18
**Weight**, *mean ± SD*, *Kg*	39.38 ± 17.96
* Female*	39.13 ± 17.19
* Male*	39.60 ± 18.68
**Height**, *mean ± SD*, *cm*	139.23 ± 24.74
* * *Female*	138.54 ± 22.71
* * *Male*	139.84 ± 26.49
**BMI**, *mean ± SD*, *Kg/m^2^*	19.06 ± 3.95
* * *Female*	19.26 ± 4.16
* * *Male*	18.89 ± 3.75
**mNF**, *mean ± SD*, *% predicted*	85.38 ± 19.47
* * *Female*	86.78 ± 18.24
* * *Male*	84.11 ± 20.51

BMI = body mass index; mNF = mean nasal flow. * all comparisons have no statistical significance.

**Table 2 jcm-11-05019-t002:** Characteristics of all the enrolled allergic patients.

Allergy Characteristics	Value
Allergics, *n* (%)	105 (35.47)
* Female*	44 (31.43)
* Male*	61 (39.10)
Allergics with at least two symptoms, *n* (%)	41 (39.05)
Allergics with at least three symptoms, *n* (%)	10 (9.52)
Allergic symptoms, *n* (%)	
* Allergic Rhinitis (AR)*	79 (75.24)
* Conjunctivitis*	20 (19.05)
* Asthma*	22 (20.95)
* Atopic dermatitis*	18 (17.14)
* Urticaria*	17 (16.19)

**Table 3 jcm-11-05019-t003:** Characteristics of COVID-19 infection.

COVID-19 Characteristics		
COVID-19 with symptoms, *n* (%)	222	(75.00)
* with more than one symptom*, *n (%)*	168	(75.68)
* Fever*	125	(56.31)
* Upper airways involvement*	96	(43.24)
* Headache*	89	(40.09)
* Anosmia*	60	(27.03)
* Dysgeusia*	46	(20.72)
* Asthenia*	42	(18.92)
* Arthromyalgia*	41	(18.47)
* Gastrointestinal symptoms*	34	(15.32)
* Dispnea*	19	(8.56)
* Skin rash*	13	(5.86)
* Pneumonia*	1	(0.45)
Disease duration, mean ± SD, day	20.09 ± 8.05	

SD = Standard Deviation.

**Table 4 jcm-11-05019-t004:** Relationship between allergies and symptomatic COVID-19.

COVID-19					
	*Asymptomatic*		*Symptomatic*		*p-Value*
	*Value*	*%*	*Value*	*%*	
**No Allergy**	55	28.8%	136	71.2%	
**Allergy**	19	18.1%	86	81.9%	
**Total**	74	25.0%	222	75.0%	0.042

The comparison between allergies and symptomatic COVID-19 (SC) highlights a higher percentage presence of SC in the allergic group.

**Table 5 jcm-11-05019-t005:** Comparison between allergies and anosmia.

No Anosmia						Anosmia				
	*Male*		*Female*		*Total*	*Male*		*Female*		*Total*
**No Allergy**	71	48.6%	75	51.4%	146	24	53.3%	21	46.7%	45
**Allergy**	56	62.2%	34	37.8%	90	5	33.3%	10	66.7%	15
**Total**	127	53.8%	109	46.2%	236	29	48.3%	31	51.7%	60

The comparison between allergies and anosmia presents a lower percentage of anosmia in the group of allergic males statistically significant (*p* = 0.007), as shown in the following Table 6.

**Table 6 jcm-11-05019-t006:** Allergy risk estimation for anosmia.

ANOSMIA			
*Risk Estimation*	*Male*	*Female*	*Total*
** *OR (CI 95%)* **	0.26 (0.09–0.73)	1.05 (0.44–2.47)	0.54 (0.28–1.00)
** *p-value* **	0.007	0.91	0.05

Allergy risk estimation for anosmia by gender.

**Table 7 jcm-11-05019-t007:** Comparison between the mean values of BMI, age, and disease duration considered in the group with and without anosmia.

*Parameter Value*	*Anosmia*		*p-Value*
	*Absent (236)*	*Present (60)*	
**BMI**, *mean ± SD, Kg/m^2^*	18.38 ± 3.48	21.76 ± 4.52	0.001
**Age**, *mean ± SD, years*	9.19 ± 4.05	12.49 ± 3.77	0.001
**Disease duration**, *mean ± SD, day*	19.33 ± 8.07	23.10 ± 9.52	0.006

Comparison between the mean values of the variables considered in two groups, with and without anosmia. BMI = body mass index.

**Table 8 jcm-11-05019-t008:** Association between anosmia and several independent variables: Logistic regression.

Variables in the Equation (Logistic Regression)							IC 95% per Exp(B)	
	B	S. E	Wald Test	gf	*p*-Value	Exp(B)	Inf.	Sup.
Gender	0.158	0.332	0.227	1	0.634	1.171	0.611	2.246
BMI	0.148	0.047	10.013	1	0.002	1.159	1.058	1.270
Allergy	0.981	0.370	7.047	1	0.008	2.667	1.293	5.504
Disease duration	0.045	0.019	5.474	1	0.019	1.046	1.007	1.085
Flow	−1.464	0.862	2.885	1	0.089	0.231	0.043	1.253
Age	0.119	0.056	4.583	1	0.032	1.126	1.010	1.256
Costant	−6.027	1.345	20.074	1	0.000	0.002		

Logistic regression model with dependent variable Y (anosmia) = 1 (yes) or 0 (not) and covariates: gender, BMI, allergies, COVID-19 disease duration, flow, and age. S. E = standard error; gf = grade of freedom; BMI = body mass index.

## Data Availability

The data presented in this study are available on request from the corresponding author. The data are not publicly available due to privacy issues.

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
