# Peer review of "COVID-19, Anosmia, and Allergies: Is There a Relationship? A Pediatric Perspective"

_jcm, 2022, doi:10.3390/jcm11175019_

Round 1

Reviewer 1 Report

 Comments to the Author

 This is a study title

 COVID-19, anosmia and Allergy: is there a relationship? A pediatric perspective.

This paper presents some  of-interest data pertaining to ansomnia after Covid-19 and relation to allergy status in 296 pediatric patients.

However, prior to consideration for publication several issues
would need to be addressed as outlined below.

A longer manuscript as presented here would need to be tightened up.

In the introduction, the aims should be clearly presented at the end of the introduction.

In methods and materials – For easier reading and understanding by the reader, please add subheadings of the procedures performed e.g. 2.1 skin prick tests  2.2 rhinomanometry 2.3 COVID -19 symphtoms 

The  sentence line 130: This test is not invasive and easy to perform and takes about 20  minutes. is not necessary

Results Informations are repeated, e.g. lines 155 and 162, in addition, the same informations are repeated in the text and tables.

Please change sentences, into impersonal (infinitive)  e.g. - As first step, we conducted a descriptive analysis of  the variables considered..   This section needs to be reworded and organized.

The Authors do not present enough detailed data for the reader to know the importance of their findings. In Tables 4 and 5, results for ansomnia and allergy are presented in all 296 patients but the study group was very diverse in age; and   how were allergies or asthma and lack of taste found in a 1 month old baby?

Discussion:  at the beginning there is no need to repeat the aims of the study, but the most important achievements of the study should be presented.

Finally, the manuscript requires rearranging and organizing.  The procedures and research tasks should be described in the same order in the methods, in the results section, and then presented and discussed in the the discussion section.

Conclusions should be clearly presented and supported by the studies presented

Reviewer 2 Report

Studies on SARSC COV2 infection in the pediatric age group are rare and extremely important in this context. The present work brings extremely important new data for all those interested in this subject. Extremely well designed, with clearly explained methodology, adequate statistics and extremely comprehensive discussions, reviewing all the news in the field in the context of their own discoveries. The authors demonstrate that allergic subjects when infected, seem to develop a form of COVID-19 symptomatic rather than asymptomatic. Furthermore, in the pediatric population, it was demonstrated that being male  and allergic seems to be a protective factor against the development of anosmia. In anosmic patients,  a statistical significant correlation betwen a high BMI, old age as well as a longer duration of COVID-19.

Author Response

Thank you very much for your comments and for showing interest in the results of our research.

Round 2

Reviewer 1 Report

Comments to the study: COVID-19, anosmia, and Allergy: is there a relationship? A pediatric perspective.

The Authors made some corrections, but the manuscript is still not clear.

Major

The Authors still have not explained how they studied anosmia in children, especially in infants? This part of the methodology is still poorly described and needs improvement. If children with AR were studied, was it asked during the interview how many children had anosmia due to rhinitis before covid -19?

 The Authors made subheadings in the method chapter but did not present the results of these subsections. What is the point of accurately describing Prick Skin Tests and rhinomanometry if the results are not presented further in the text?

Conclusions are still not clearly presented, for example,  the authors write that controlled allergy is not a risk factor for infection but in the materials,  methods and results there are no data on the control of allergic diseases in patients. In addition, a group of 296 patients is not representative for determining risk factors. Similarly, it is difficult to determine the protective role of sex in anosmia only based on statistical analysis without proving cause-and-effect relationships.

Line 530    Allergies, if under control, did not represent risk factors for SARS CoV2 infection.

Line     532   Furthermore, in the pediatric population, considering anosmia which, which is one of the cardinal symptoms of COVID-19, it was seen that being male and allergic seems to be a protective factor against the development of this symptom during infection.

This sentence is the result, not the conclusion:

Line 535 Then looking at anosmic patients, we found a statistical significant correlation with a higher BMI, an older age as well as a longer duration of  COVID-19 in the acute phase.

The abstract needs to be changed according to the background, materials, methods, results, and, conclusions sections.

Minor

Still, English needs proofreading.

Please specify inaccuracies: anosmia was presented by  27.03% of patients in tab1, while in the text by-20.03%.

Author Response

see the uploaded file, please
